# Influence of the Iron as a Dopant on the Refractive Index of WO_3_

**DOI:** 10.3390/ma14195845

**Published:** 2021-10-06

**Authors:** Mariana Osiac, Iulian Boerasu, Madalin-Stefan Radu, Maria Jigau, Ion Tirca

**Affiliations:** 1Department of Physics, Faculty of Sciences, University of Craiova, 200585 Craiova, Romania; mariana71osiac@gmail.com (M.O.); madalin.radu17@gmail.com (M.-S.R.); mariajigau@mail.ru (M.J.); 2Department of Lasers, Plasma and Radiation Physics, National Institute for Lasers, 077125 Magurele, Romania; iulian.boerasu@inflpr.ro

**Keywords:** thin film, pure and 2% Fe-doped tungsten oxide, refractive index, optical band gap

## Abstract

Results on studies of pure tungsten oxide WO_3_ and 2, 3 and 4% Fe-doped WO_3_ grown on the sapphire substrates by reactive pulsed laser deposition technique are reported. From X-ray diffraction it results that the crystalline structures changed with the substrate temperature and the peaks diffraction having a small shift by the amount of Fe content in WO_3_ lattice was noticed. Scanning electron microscopy presented a random behavior of WO_3_ nanocrystallites size with substrate temperatures. In the presence of 2% Fe-doped WO_3_, the nanocrystallites size varied gradually from 60 nm to 190 nm as substrate temperature increased. The transmission spectra of the pure and 2, 3 and 4% Fe-doped WO_3_ films were obtained within the 300–1200 nm spectral range. The refractive index of WO_3_ and Fe-doped WO_3_ layers were calculated by the Swanepoel method. The refractive index of pure WO_3_ shows a variation from 2.35–1.90 and for 2% Fe-doped WO_3_ from 2.30–2.00, as the substrate temperature increased. The contents of 3 and 4% Fe-doped WO_3_ presented nearly identical values of the refractive index with pure and 2% Fe-doped WO_3_, in error limits, at 600 °C. The optical band gap changes with substrate temperature from 3.2 eV to 2.9 eV for pure WO_3_ and has a small variation with the Fe.

## 1. Introduction

Tungsten trioxide (WO_3_) is an important n-type semiconductor with excellent properties for electrochromic devices, gas sensing, heterogeneous catalysts, chromogenic devices, solar energy devices, field electron emission, etc.

The sensors based on WO_3_ are highly sensitive for very low concentrations of toxic gases such as NO_x_, H_2_S, NH_3_ and CO [1,2,3,4,5]. Moreover, WO_3_ films with transparent characteristics in the visible region are of interest in electrochromic phenomena for applications in display devices or smart windows [6]. The sub-stoichiometric intermediate phases make WO_3_ appropriate for electronic properties such as superconductivity and ability to carry a charge. The WO_3_ metal oxide nanowires have a great importance because of their distinctive electrochromic, optochromic, gas chromic and magnetic properties. Thus WO_3_ metal oxide nanowires have become good candidates for lithium-ion batteries, catalysts, electrochromic devices and sensors. The metal oxide nanowires provide a unique system for understanding superconductivity and colossal magnetoresistance properties [7,8,9,10], therefore WO_3_ nanowires have been produced by different methods [11]. Regarding the sensor applications, some limitations appear, such as high operating temperature, slow response time, and poor selectivity and reproducibility, making them impractical monitors for area air quality and safety. To overcome these limitations, research attention has been focused on increasing the sensor performance, which is usually realized by the incorporation of metals in the base metal oxide sensors.

With respect to other deposition techniques, pulsed laser deposition (PLD) allows control of the layer properties by adjusting various parameters such as laser fluence, substrate temperature, or gas pressure. The growth of WO_3_ thin films by PLD in an oxygen atmosphere was firstly reported by Poniatowski et al. [12]. Deposition of pure and doped WO_3_ layers with noble metals has been mainly done by the PLD method [13,14,15,16,17].

The WO_3_ thin films and nanostructures used for technological applications need a study of the optical properties such as refractive index and optical band gap. The optical characterization of thin films can be determined by the Swanepoel method [18]. This method uses the envelopes on the interference fringes obtained by the transmission spectra of films grown on thick transparent substrates [19,20,21]. The two-term Cauchy dispersion function was used to fit the refractive index obtained from the transmission spectra. The transmission spectra was done in the 300–1200 nm wavelength range. By means of the Swanepoel method the refractive index has been calculated taking into account the values of *T_max_* (*λ*) and *T_min_* (*λ*) of the transmission spectra.

The choice of the range 300–1200 nm is done by the fact that above the 1200 nm the maximum and minimum peaks become very rare and wide and leads to refractive index values far outside the error limit and implicitly to a totally wrong approximation.

In order to improve the optical properties of WO_3_ layer, several concentrations of the dopant (2, 3 and 4% Fe) were chosen and investigated at various temperatures. It was found that the concentration of 2% Fe was close to pure WO_3_ crystalline structure. 

The goal of this paper is to measure refractive indexes in the 300–1200 nm spectral range of pure and 2, 3, 4% (weight) Fe-doped WO_3_ thin films deposited by PLD on the sapphire substrate by using the Swanepoel method. The refractive index was obtained for layers grown at various temperatures in the range 600–700 °C at 4 h deposition time. A comparison among the refractive indexes of pure and 2% Fe-doped WO_3_ layers was done. The optical properties such as optical band gap have been measured. The relationship between the crystalline structure and the morphology of the film surface has been investigated. The optical parameters, such as refractive index (n), film thickness (d), or optical band gap (Eg) for the films prepared at different temperatures are obtained from the corresponding transmission spectra.

## 2. Material and Methods

### Thin Film Investigation Methods

The pure and 2, 3 and 4% Fe-doped WO_3_ thin films were grown on sapphire substrates by PLD [10,11,12,13,14,15]. An Nd-YAG laser (266 nm laser wavelength) with a repetition rate of 10 Hz and a pulse duration of 10 ns was used. The diameter of the laser beam is approximately 1 mm and the energy was adjusted to obtain a fluence of approximately 1.7 J/cm^2^. The sapphire substrates were located at 4.5 cm from the targets (WO_3_ and 2, 3, 4% Fe + WO_3_) and were radiative heated between 600–700 °C, with a heating rate of 15 °C/min. The deposition has been carried out in oxygen atmosphere, at a pressure of 120 mTorr. 

A model 6000 X-ray diffraction system (Shimadzu, Tokyo, Japan) was used to investigate the crystalline structure of the pure and 2, 3 and 4% Fe-doped WO_3_ samples.

A field emission scanning electron microscope (FE-SEM; Thermo Fisher Scientific, Waltham, MA, USA) coupled with a EDX (model ApreoSLoVachas, Hillboro, OR, USA) was used to study the surface morphology of samples. The plane view (0° tilt) samples surface and cross-section (90° tilt) morphology have been investigated in high-vacuum operation mode (6.1 × 10^−4^ Pa) at a working distance of 10.5 mm, using 10 kV as accelerating voltage and a nominal electron beam current of 13 Pa. The SEM images were recorded on as-processed samples, i.e., without any conductive coatings. 

The optical properties of pure and 2, 3, 4% Fe doped WO_3_ films have been investigated by the Swanepoel method. The transmission spectra within the 200–1200 nm spectral range were recorded using a Specord 210 plus instrument (Analytik Jena AG, Jena, Germany) in scanning mode. The spectrophotometer was coupled with an integration sphere that ensures uniform illumination at a normal incidence of the light on the film surface (1 mm), independent of the scattering or the deflection beam. The optical transmission measurements were done in the center of the samples. The oscillations within the transmission spectra indicate a source of optical interference. These oscillations are an effect of the pure and doped WO_3_ thin film and substrate. Applying the Swanepoel method the refractive index value of crystalline films was obtained. The Swanepoel method described briefly here takes into account the mathematical functions that envelope the transmittance spectra, which include multiple maxima and minima due to interference fringes. The sapphire substrates show a refractive index s, given by Equation (1):(1)$$s=\frac{1}{T_{S}}\;+\;{\left(\frac{1}{T_{S}}-1\right)}^{1/2}$$
where $T_{s}$ is the transmittance of the sapphire relative to air. 

The refractive index of pure and 2, 3 and 4% Fe-doped WO_3_ in the weak and medium absorption has been calculated taking into account the values of $T_{max}\left(\lambda \right)$ and $T_{min}\left(\lambda \right)$ from the transmittance spectra. The values of the refractive index of the pure and doped WO_3_ layers, in the first approximation, denoted by $n_{1}$, is calculated by the following expression:(2)$$n_{1}={\left(N_{1}\;+\;{\left(N_{1}^{2}-s^{2}\right)}^{1/2}\right)}^{1/2}$$
where $N_{1}$ is a parameter depending on the $T_{max}$ and $T_{min}$ and substrate thickness, s, from Equation (1). Then film refractive index value $n_{1}$ obtained by Equation (2) is improved after calculating the film thickness, $d_{1}$. By means of the basic equation of interference fringes, the film thickness is calculated as:(3)$$d_{1}=\frac{\lambda_{1}\lambda_{2}}{2\left(\lambda_{1}n_{e2}-\lambda_{2}n_{e1}\right)}$$
with $n_{e1}$ and $n_{e2}$ the refractive indices at two successive maxima or minima corresponding to the wavelengths, $\lambda_{1}$ and $\lambda_{2}$, respectively. The interference fringes equation is $2nd=m_{0}\lambda$, where $m_{0}$ is the first approximation of m, which is an integer for maxima and a half-integer for minima. The exact value of m increases the film thickness accuracy. The new values of the thickness $d_{2}$ are derived from Equation (3). 

The final value of $d_{2}$ is found after calculation of $n_{1}$ by Equation (2). The film thickness accuracy is in the range of 1.1–3%. The refractive index values of pure and Fe doped WO_3_ films are calculated using the exact value m and the thickness,$\;d_{2}$. Finally, the refractive index values are fitted by a two-term Cauchy dispersion relation in the range of measurements:(4)$$n\left(\lambda \right)=B\;+\;\frac{C}{\lambda^{2}}$$
where, B and C are constants. In the calculation of $\;d_{2}$ higher values of the layer thickness are not considered.

The optical band gap $E_{g}$ has been determined from absorption coefficient data as a function of photon energy using the Tauc relation:(5)$${\left(\alpha h\nu \right)}^{1/2}=A\;\left(h\nu -E_{g}\right)$$
where  A is the slope of the Tauc edge, named band parameter, which might depends on the width of localized states in the bandgap. The ordinate is given by ${\left(\alpha h\nu \right)}^{1/r}$, where α is the absorption coefficient, the exponent 1/r denotes the nature of the transition and r may take the values ½, 3/2, 2, or 3, depending on the transitions being either direct allowed, direct forbidden, indirect allowed, or indirect forbidden, respectively. In the present case, the exponent is ½ corresponds to the value r=2, being an indirect allowed transition.

## 3. Results and Discussions

### 3.1. X-ray Measurements

Figure 1a,b show the X-ray spectra for all samples in the 2θ = 21–52° range. The samples present the γ-monoclinic crystalline structure (ICDD card No.043-1035) of the WO_3_ layer. The X-ray spectra of the layers deposited at temperatures 600, 660, 680 and 700 °C respectively (Figure 1a) presented diffraction peaks at (002), (022) and (200), with a preferential crystalline plane (002) along the c-axis. The intensities of the diffraction peaks (002), (022) for the sample grown at 680 °C are nearly identical.

Figure 1b illustrates the X-rays diffraction peaks for WO_3_ layers in the 2θ = 45–52° range. The X-rays diffractions peaks are (004), (040), and (400), respectively. The layer grown at 660 °C has the highest intensities of the (004) diffraction peak. The intensities of the diffraction peaks are gradually increasing with temperatures an exception being at 660 °C. In the 640–660 °C range, the crystalline structure of the WO_3_ has the diffraction peaks intensities nearly identical, thus a random orientation of the monoclinic structure is presented. 

Figure 2a–c present the X-ray spectra for all samples in the 2θ = 20–52° range. The figures evidence the crystalline structure of 2% Fe-doped WO_3_ layer. The X-ray spectra of the layers grown at temperatures 600–700 °C (Figure 2a,b) exhibit characteristic diffraction peaks at (002), (022) with a preferential crystalline plane (022), i.e., b-axis. 

In the XRD pattern of Figure 2a,b the intensity of peaks corresponding to (022) increases gradually at 600, 650 and 680 °C, respectively and strongly decreases at 700 °C. A gradual decreasing of the diffraction peaks intensities with increasing temperature is shown in Figure 2c.

In Figure 2a, the diffraction peak (200) of the layer grown at 600 °C vanishes, this effect being related to the small amount of Fe doped in the host lattice of WO_3_. 

The effect of Fe as a dopant is visible for 3% and 4%Fe doped WO_3_ layers. The diffraction peaks (022), (200), (040) and (400), respectively for 3 and 4% Fe doped WO_3_ layers vanished. To analyze the effect of Fe doped WO_3_ layer, the plots are made as a function of temperature for pure/doped WO_3_ for a better characterization, as seen in Figure 3a,b.

Figure 3a presents a small deviation of (200)/(022) diffraction peaks compared to pure WO_3_ with increasing the content of (2, 3, 4%) Fe-doped WO_3_. A small deviation of diffraction peaks (004)/(040) of Fe-doped WO_3_ compare to pure WO_3_ is noticed in Figure 3b. This deviation is attributed to the ionic radius of octahedral coordinated Fe^3+^ of 0.64 Å, close to that of W^6+^ with identical coordination number (0.60 Å) [22,23]. The crystalline field stabilization energy of octahedral coordination for Fe^3+^ ion is higher than that of tetrahedral coordination. Thus, Fe^3+^ ions could enter in the host lattice of WO_3_ and replace W^6+^ ions. The difference of the ionic radius of octahedral coordinated (0.04 Å) results in a small shift in peak positions observed in Figure 3a,b. The observed shift in peak positions of Fe-doped WO_3_ compared to pure WO_3_ can be attributed to the small difference between of W^6+^ and Fe^3+^ ionic radii. The Fe^3+^ ionic radius being a little greater than that of W^6+^, a slight distortion appeared in the crystal lattice when WO_3_ is doped with Fe, resulting in a shift in the diffraction peaks. The lattice distortions produce numerous defects in the film.

### 3.2. SEM Measurements

The top-view high magnification secondary electron (SE) images of pure WO_3_ and 2% Fe doped WO_3_ thin films as function of deposition temperature are shown in Figure 4 and Figure 5. 

According with the recorded images, the impact of the substrate temperature on the surface morphology of processed samples is important. Moreover, analyzing the recorded images, the surface morphology evolution seems having a random dependence on the substrate temperature increasing. Indeed, a rough nanostructured surface consisting in well-defined surface grain features illustrated the morphology of pure WO_3_ grown at 600 °C (Figure 1a). The observed WO_3_ grains are like pyramid-shaped, sharp grains with a size of approximately 130 nm and micro-holes are visible around the grains. A fine microstructure and uniform distribution of the particles are evident in the micrographs recorded on the pure WO_3_ films grown at 660 °C (Figure 1b). The grains are more homogeneous and compactly arranged than the grains observed in WO_3_ films grown at 600 °C. The average size of surface grains of those films is decreased to approximately 73 nm. Even low as volume fraction, micro-holes are observed preferentially located on the gain-boundaries. Further increasing the deposition temperature at 680 °C, the surface morphology of the growth films consists in grains of pyramidal shape, with a size of about 127 nm, randomly oriented with respect to the substrate surface. A large amount of physical defects, like micro-holes, are visible around the grains. Particularly, a surface morphology featured by grain size decreasing (average grain size of 97 nm), and rough rectangular-shaped grains were observed in the film grown at a substrate temperature of 700 °C. However, for those WO_3_ films, several nano-sized holes are distributed both on the grains, and at the grain boundaries.

The high magnification SEM images of 2% Fe-doped WO_3_ films depending on various temperatures are shown in Figure 5a–d.

The effect of temperature on the surface morphology of Fe-doped WO_3_ films is remarkable. The SEM data of 2% Fe-doped WO_3_ films can be divided into three temperature areas where the morphology differences are significant. The first area contains the 2% Fe- doped WO_3_ films grown at temperatures of 600 °C and 650 °C, respectively. In this temperature area, the morphology of the deposited films has a “cobbled” appearance, with well-defined grains of an average size of about 60 nm. The second area of 2% Fe-doped WO_3_ films grown at temperatures of 650 °C and 680 °C, with a rough surface morphology, composed of slightly larger pyramid-shaped crystallites (average size of 90 nm) is observed. However, a large amount of physical defects like microporosity is obviously visible in the grains boundaries area. The third temperature area belongs to films grown at 700 °C. According to the recorded images, at 700 °C deposition temperature leads to a simultaneous increase of the size of the grain surface, and grain size homogeneity. The films grown at this temperature show a compact surface morphology formed by large pyramidal-shaped grains, with an average size of 194 nm. Nano-porosity located on the grains surface is visible in the recorded images. 

### 3.3. Refractive Index

The calculation of the refractive index from Equations (2) and (3) using the Swanepoel method are displayed in Table 1 and Table 2 for two extreme temperatures (600 and 700 °C, respectively), particularly for the WO_3_ + 2% Fe layer. The layer thicknesses with larger values are removed from the final calculation of d_2 as is shown in Table 1 and Table 2. The layer thickness calculated by the Swanepoel method and the thickness measured by SEM cross-section, in Table 1 and Table 2 show good agreement. The refractive indexes and layer thickness for sample grown at 3 and 4% Fe-doped WO_3_ are calculated in the similar way as for the samples with 2% Fe + WO_3_. 

The graphs corresponding to the refractive index values n_2_ are presented in Figure 6a,b for the samples grown at 4 h substrate temperature in the range of 600 and 700 °C, respectively. The refractive index values obtained from the transmission spectra are fitted using a two-term Cauchy dispersion function in the wavelength range of 400–1200 nm. 

In Table 1 and Table 2, the final results of thickness, d_2_, the refractive index, and the Cauchy dispersion are presented. The thickness values of d_2_ are derived from Equation (3) together with n_1_ calculated by Equation (2), and finally the values of d_2_ are obtained with good accuracy.

The values of the refractive index from Figure 6a,b are varied from 2.32–2.00 at 600 °C and 2.20–1.90 at 700 °C, respectively. The similar values of refractive index in this temperature range have been reported previously [24,25]. Considering the present work and the ref. [24], the refractive index values seem to be independent of the transparent substrates such as quartz and sapphire.

The refractive index of 2% Fe-doped WO_3_ layer grown at 4 h deposition time at 600 °C presents values in the range 2.40–2.04, as shown in Figure 7a. The tiny amount of 2% Fe presents in the WO_3_ layer does not significantly the refractive index change compared with pure WO_3_. Figure 7b presents the variation value of the refractive index, 2.32–2.00, as the temperature increased to 700 °C.

To understand the variation of the refractive index as the temperature increased, the plots of the refractive indexes are presented below, in Figure 8a,b. The refractive index presented in Figure 8a corresponds to a pure WO_3_ layer, while that in Figure 8b corresponds to a 2% Fe-doped WO_3_ layer, at temperatures in the 600–700 °C range. 

In the Figure 8a the refractive index of pure WO_3_ varied in the range of 2.40–2.14 for temperatures 600, 640, 660 and 680 °C, respectively. At a high temperature of 700 °C the refractive index has values from 2.16–1.92. A similar observation was found in [24,25,26,27]. This lower value of the refractive index could be explained by the self-annealing process that is activated at higher deposition time and substrate temperatures, the presence of the vacancies created in an atmosphere of oxygen during deposition. From Figure 8b, the variation of refractive index displayed values of 2.4–2.04 as the temperature increased. In the range of 600–700 °C, the refractive index values seems to have an approximately identical value within the error limits as the temperature increased. The refractive index of the layer grown at 640 °C has a high value, and it was supposed that around at 600–640 °C, at this critical temperature a change in the crystalline structure appeared. The X-ray measurements presented a random orientation of the monoclinic structure of the layer grown at 640 °C. In the hypothesized that since Fe has an atomic radius of 0.64 nm close to the size of W (0.62 nm), this can be introduced in a small quantities as a substitutional impurity in the WO_3_ crystals causing a distortion of the crystal structure. Near to the oxygen vacancies considered in the WO_3_ layer under high temperature, similar defects of the WO_3_ lattice generated by Fe doping influenced the refractive index at 640 °C. Therefore, the presence of Fe in the host lattice of WO_3_ could produce a mismatch of the lattice changing the properties of the film as well the type of strain during the growing process. It is possible that at high temperature some formed Fe oxide bonds are broken since the layer surface is under the influence of the energetic species of the plasma. 

To observe the effect of Fe-doped WO_3_ on the refractive index, the measurements of 3% and 4% Fe-doped WO_3_ were performed at 600 °C. Figure 9 show the refractive index of pure and Fe-doped WO_3_.

From Figure 9, the refractive indexes for pure and 2% Fe-doped WO_3_ seem to be nearly identical within the error limit, since these Fe percentages are considered smaller impurities in the WO_3_ lattice. The refractive index of the 3% Fe-doped WO_3_ is slightly smaller. It is worth mentioning that in case of the sapphire substrate, a few interference fringes exist in the transmittance spectra, thus the errors bars are high. Therefore, a decrease in the refractive index at 3% Fe in the host lattice of WO_3_ is visible. The refractive index of 4% Fe-doped WO_3_ is rising above the pure WO_3_. A first decrease in the refractive index and then an increase has been noticed for chalcogenide glasses with variation of Te content. As an explanation, such a variation of the refractive index was attributed to a large number of defects, Te bindings in chalcogenide and the alteration of the valence band [27]. The changes in the refractive indexes of 3 and 4% Fe doped in WO_3_ are strongly related to the modification of crystalline structure observed in X-ray measurements (Figure 3a,b) and a few diffraction peaks vanished. Unlike 2% Fe doping, an increase in the content of Fe, defects, mismatches and distortions of the lattice became noticeable. Moreover, the defects of the WO_3_ lattice lead to a sub-stoichiometry and internal strain in the grown layer. The small amount of Fe changes the internal structure of the host lattice, leading to formation of a composition with a degree of disorder and higher localized state densities. 

In Figure 10, the EDX measurements show the elemental species contained in the WO_3_ + 2% Fe layers at 600 °C and 680 °C, respectively. The temperature of 600 °C presents an amount of 0.23% Fe in the layer still smaller than it is in the target. The amount of Fe in the layer is shown in Table 3. It could be concluded that EDX measurements are not suitable for determination of the Fe content in these thin films.

It is worth mentioning that in the case of a sapphire substrate it is possible that the little amount of the oxygen could be slightly higher since a part of it is coming from the substrate.

### 3.4. Optical Band Gap Measurements

The plots of ${\left(\alpha h\nu \right)}^{1/2}\;$ versus hν, by Equation (4), for the WO_3_ samples grown at 4 h deposition time at two temperatures (600 and 700 °C, respectively) are presented in Figure 11a,b.

The energy band gap of WO_3_ layer is 3.2 eV at 600 °C, while as the temperature increases to 700 °C the energy band gap becomes 2.8 eV. This decrease in the energy is usual, since as the temperature increase the energy band gap became smaller.

Figure 12a–d show the energy band gap for Fe-doped WO_3_. In Figure 12a,b, the energy band gap for 2% doped WO_3_ at 600 and 700 °C respectively, were presented. The energy band gap at 600 °C is 3.18 eV while at 700 °C it is 2.92 eV. Unlike pure WO_3_ the energy band gap of 2% Fe-doped WO_3_ has a small variation of approximately 0.02 eV at 600 °C while at 700 °C it is nearly identical.

In Figure 12c,d the energy band gap of 3 and 4% Fe-doped WO_3_ at 600 °C is presented. The small change observed for pure and Fe-doped WO_3_ is mainly caused by the amount of Fe in the WO_3_ host lattice, leading to a small decrease in the value of the band gap. The optical band gap is strongly dependent on the amount of Fe atoms that have the tendency to enter in the lattice producing a physical disordering and creating localized states in the forbidden gap leading to a lower optical band gap. Therefore, a large number of charged defect states, induces structural changes in the host network, as it was observed in X-ray and SEM measurements and in the variation of the refractive index with temperature and amount of Fe. The increase of the Fe content leads to the formation of the composition and the densities of the localized states, so the small amount of Fe acts as an impurity center in the mobility gap. The distribution of the localized states is modified and new states are created in the mobility gap that change the optical band gap.

## 4. Conclusions

The refractive index of pure and 2, 3 and 4% Fe-doped WO_3_ layers were calculated from the transmittance spectra by the Swanepoel method. The crystalline structure of 2, 3 and 4% Fe-doped WO_3_ showed a small shift compared with pure WO_3_. The diffraction peaks at (022), (200), (040) and (400) vanished with the increase of Fe content in the WO_3_ host lattice at 600 °C. The nanocrystallite size measured from SEM images of the pure WO_3_ displayed a random variation with the increase of substrate temperature. Increasing the substrate temperature to 680 °C, the surface morphology of the grown films consists of pyramidal-shape grains, randomly oriented with respect to the substrate surface. A large amount of physical defects was observed, like micro-holes, visible around the grains as shown by SEM. Particularly at 700 °C, a surface morphology with rectangular-shaped grains was observed. The 2% Fe-doped WO_3_ presented an increase of the nanocrystallite size from 60 nm–197 nm as the substrate temperature increased between 600–700 °C. The morphology of the films has a “cobbled” appearance, with well-defined grains at 600 °C. The films grown at temperatures of 650 °C and 680 °C had a rough surface morphology, composed of slightly larger pyramid-shaped crystallites and a large amount of physical defects like micro-porosity visible at the grain boundaries site. A 700 °C temperature leads to a compact surface morphology formed by large pyramidal-shaped grains and nano-porosity located on the grains’ surface.

The refractive index of pure WO_3_ at temperatures 600, 660, 660 and 680 °C, respectively had the identical values ranging from 2.4–2.04 while for 700 °C it decreases to 2.16–1.92 in the 400–1100 nm wavelength range. The value refractive index of 2% Fe-doped WO_3_ was 2.3–2.14 at 650 °C and slightly smaller (2.3–2.0) at 660, 680 and 700 °C, respectively. An increase to 3 and 4% Fe content in the WO_3_ lattice led to nearly identical values of the refractive index as pure and 2% Fe-doped WO_3_ at 600 °C. It was obvious that the 3% Fe content is slightly lower than 2% Fe-doped WO_3_ while at 4% Fe it is slightly higher than pure WO_3_. Therefore, we presumed that the refractive index with Fe content variation could be considered nearly identical within the error limits at 600 °C. The modification of the refractive index with different Fe content in the WO_3_ lattice was in agreement with the changes in the crystalline structure observed by X-ray measurements.

The optical band gap varied with temperature for pure WO_3_ from 3.2 eV at 600 °C and 2.8 eV at 700 °C, respectively. The 2, 3 and 4% Fe-doped WO_3_ presented a small variation of optical band gap at 600 °C.

## Figures and Tables

**Figure 1 materials-14-05845-f001:**
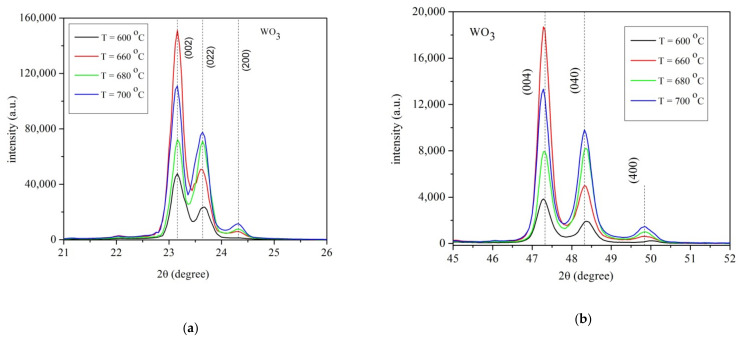
X-ray patterns for WO_3_ layers grown on sapphire substrate at temperatures T = 600 °C, T = 660 °C, T = 680 °C, T = 700 °C and deposition time 4 h.(**a**) range 2θ = 21–26°; (**b**) range 2θ = 45–52°.

**Figure 2 materials-14-05845-f002:**
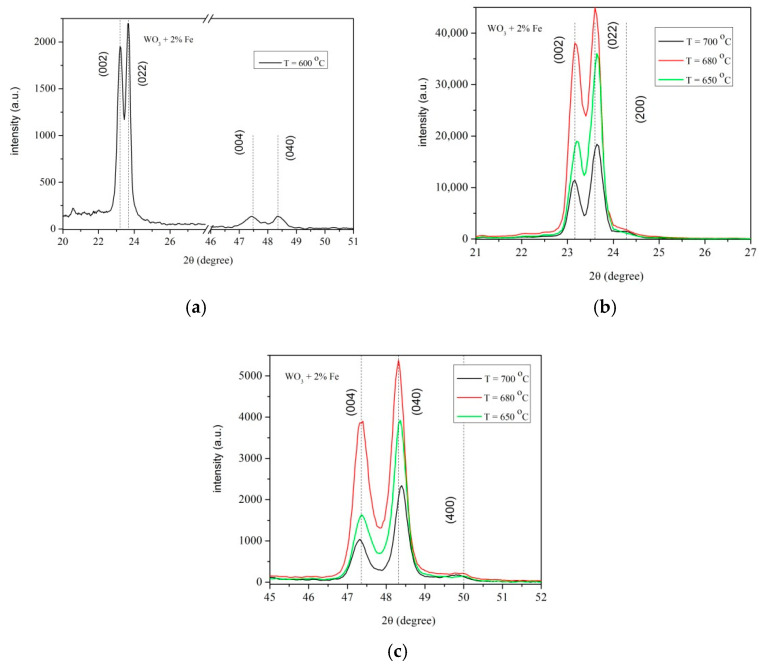
(**a**–**c**) X-ray patterns of 2%Fe doped WO_3_ layers grown on sapphire substrate at various temperatures and 4 h deposition time (**a**) T = 600 °C in the range 2θ = 20–51°; (**b**) T = 650 °C, T = 680 °C, T = 700 °C in the range 2θ = 21–7°; (**c**) T = 650 °C, T = 680 °C, T = 700 °C in the range 2θ = 45–52°.

**Figure 3 materials-14-05845-f003:**
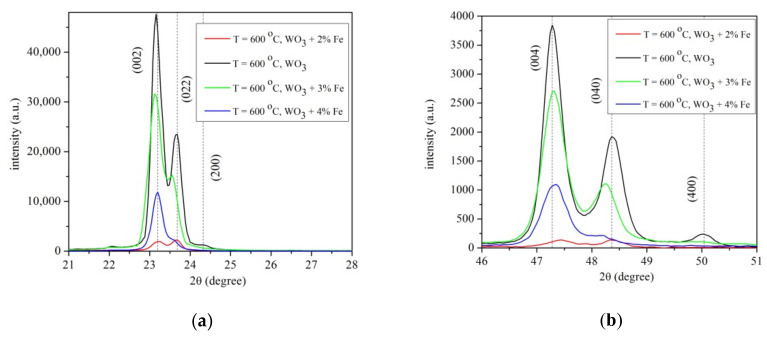
(**a**,**b**) X-ray pattern of pure and 2, 3 and 4% Fe doped WO_3_ layers grown on sapphire substrate at 4 h deposition time and substrate temperature T = 600 °C. (**a**) pure and 2, 3 and 4% Fe doped WO_3_ in the range 2θ = 21–28°; (**b**) pure and 2, 3 and 4% Fe doped WO_3_ in the range 2θ = 46–51°.

**Figure 4 materials-14-05845-f004:**
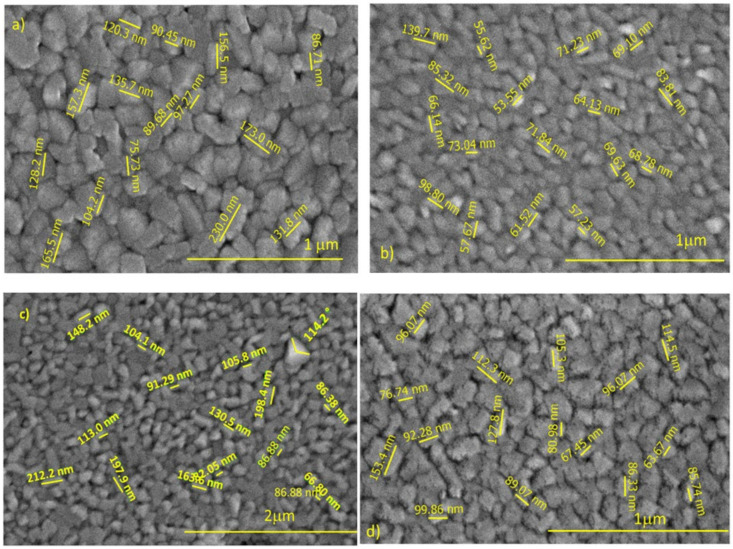
SEM image of the WO_3_ layer grown at 4 h deposition time and various temperatures. (**a**) T = 600 °C; (**b**) T = 660 °C; (**c**) T = 680 °C and (**d**) T = 700 °C.

**Figure 5 materials-14-05845-f005:**
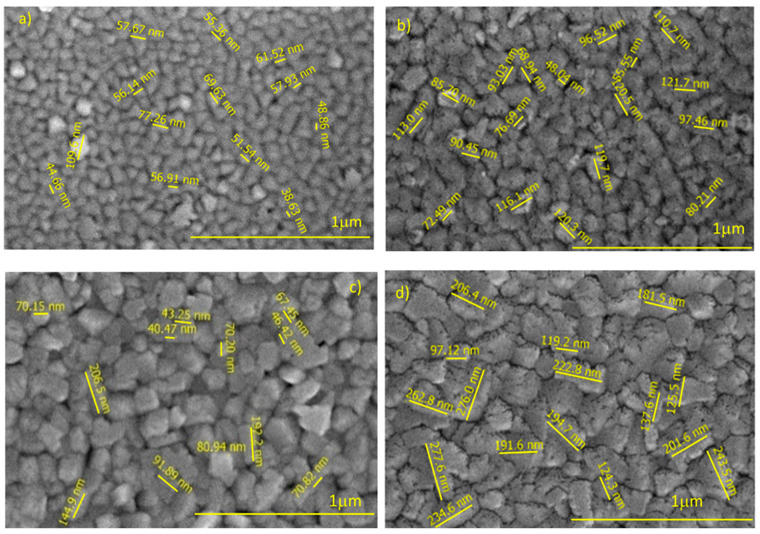
SEM image of the 2%Fe doped WO_3_ layer grown at 4 h deposition time and various temperatures. (**a**) T = 600 °C; (**b**) T = 650 °C; (**c**) T = 680 °C and (**d**) T = 700 °C.

**Figure 6 materials-14-05845-f006:**
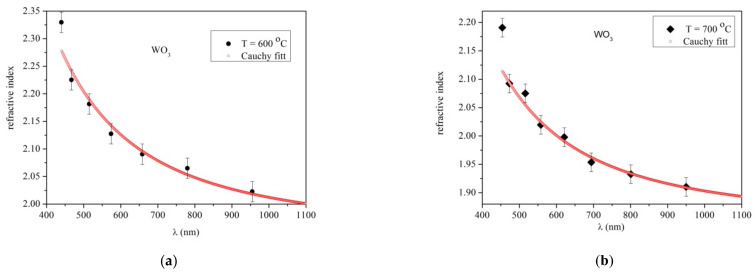
Refractive index plots of pure WO_3_ layer grown at 4 h deposition time. (**a**) T = 600 °C; (**b**) T = 700 °C.

**Figure 7 materials-14-05845-f007:**
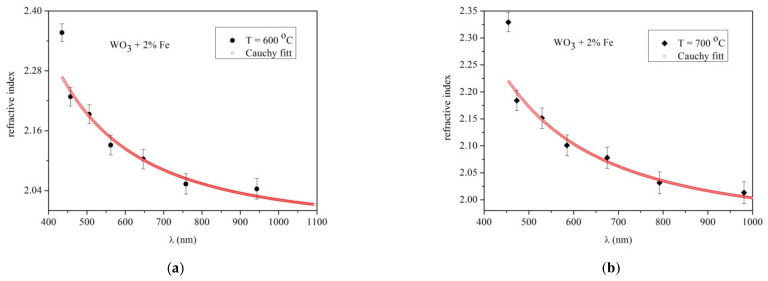
Refractive index plots for 2% Fe doped WO_3_ layer grown at 4 h deposition time. (**a**) T = 600 °C; (**b**) T = 700 °C.

**Figure 8 materials-14-05845-f008:**
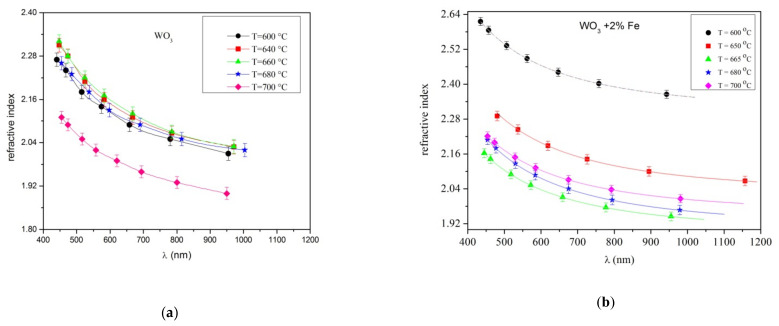
The refractive index depending on the substrate temperatures. (**a**) pure WO_3_ layer; (**b**) 2% Fe doped WO_3_ layer.

**Figure 9 materials-14-05845-f009:**
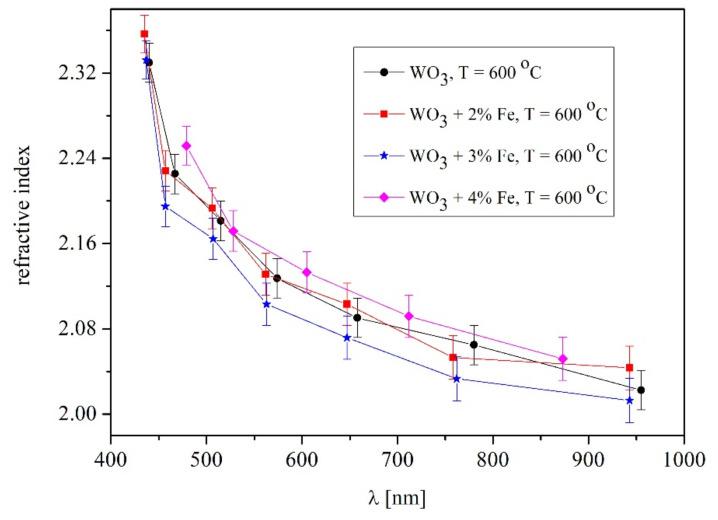
The refractive index of pure and Fe-doped WO_3_ layers grown at 600 °C. The refractive index of pure WO_3_ and 2, 3 and 4% Fe-doped WO_3_ are compared.

**Figure 10 materials-14-05845-f010:**
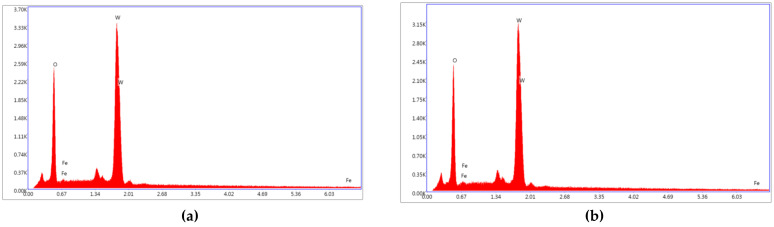
The EDX measurements for WO_3_ + 2% Fe grown at 4 h deposition time. (**a**) T = 600 °C; (**b**) T = 680 °C.

**Figure 11 materials-14-05845-f011:**
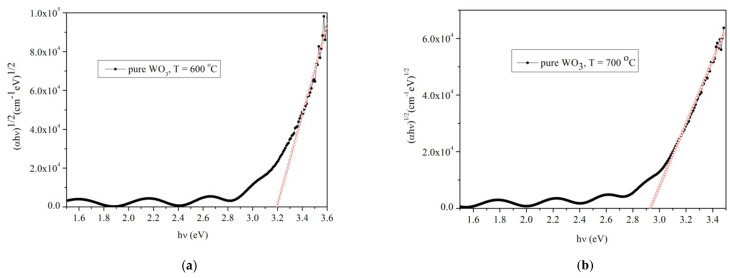
Optical band gap of pure WO_3_ layers grown on sapphire substrates: (**a**) sample at 600 °C (**b**) sample at 700 °C.

**Figure 12 materials-14-05845-f012:**
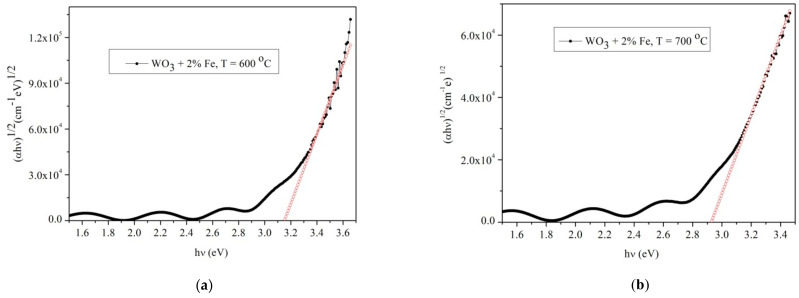
Optical band gap of pure WO_3_ layers grown on sapphire substrates:(**a**) 2% Fe-doped WO_3_ at 600 °C; (**b**) 2% Fe- doped WO_3_ at 700 °C; (**c**) 3% Fe-doped WO_3_ at 600 °C; (**d**) 4% Fe-doped WO_3_ at 600 °C.

**Table 1 materials-14-05845-t001:** The parameters obtained for WO_3_ + 2% Fe layer samples grown at T = 600 °C and 4 h deposition time.

WO_3_ + 2% Fe onto Sapphire Substrate, T = 600 °C, 4 h Deposition Time										
λ	$T_{M}\;$	$T_{m}\;$	s	$n_{1}$	$d_{1}\;$	$m_{0}\;$	m	$d_{2}\;$	$n_{2}\;$	$n_{2}$ Cauchy Dispersion
435	0.782	0.63	1.708	2.34		5.55	5	463.95	2.35	2.26
457	0.831	0.73	1.704	2.09	820.76	4.71	4.5	491.61	2.22	2.24
506	0.968	0.76	1.685	2.28	500.75	4.64	4	443.84	2.19	2.18
562	0.97	0.803	1.679	2.15	699.78	3.95	3.5	455.93	2.13	2.14
647	0.98	0.812	1.672	2.14	529.54	3.41	3	452.50	2.10	2.10
758	0.992	0.824	1.667	2.12	642.9	2.89	2.5	444.83	2.05	2.06
943	0.993	0.882	1.663	1.97		2.15	2	477.54	2.04	2.02
$d_{1}=515.15\;nm\pm 14.39\;nm\;\;\;\;\delta =2.8\%$										
$d_{2}=461.45\;nm\pm 6.67\;nm\;\;\;\delta =1.44\%$										
SEM d=418.16 nm±17.28 nm δ=0.09%,										
$n\left(\lambda \right)=1.964+\frac{0.5752*10^{5}}{\lambda^{2}}$										

Note: The underlined values in the table are attached in this way to indicate that they are obtained from the transmission graph obtained by spectrophotometry. The other values are obtained from the envelope curves according to Swanepoel theory. The numbers in the table are cut to show that their values obtained from the experiment were not used in the calculation of the final value of n_2_ because they had very large deviations, well above the error limit.

**Table 2 materials-14-05845-t002:** The parameters obtained for WO_3_ + 2% Fe layer sample grown at T = 700 °C and 4 h deposition time.

WO_3_ + 2% Fe onto Sapphire Substrate, T = 700 °C, 4 h Deposition Time										
λ	$T_{M}\;$	$T_{m}\;$	s	$n_{1}$	$d_{1}\;$	$m_{0}\;$	m	$d_{2}\;$	$n_{2}\;$	$n_{2}$ Cauchy Dispersion
454	0.737	0.589	1.7	2.38		5.50	5	475.33	2.32	2.22
473	0.807	0.719	1.69	2.04	414.4	4.53	4.5	519.2	2.18	2.19
529	0.91	0.766	1.68	2.14	637.27	4.24	4	493.47	2.17	2.14
585	0.941	0.808	1.67	2.07	548.88	3.71	3.5	493.1	2.10	2.11
675	0.977	0.818	1.67	2.12	531.17	3.29	3	477.41	2.07	2.07
792	0.977	0.835	1.66	2.06	486.16	2.73	2.5	479.35	2.03	2.03
981	0.988	0.84	1.66	2.07		2.21	2	473.15	2.01	2.00
$d_{1}=523.57\;nm\pm 6.36\;nm\;\;\;\;\delta =7.01\%$										
$d_{2}=487.28\;nm\pm 6.16\;nm\;\;\;\delta =1.26\%$										
SEM d=456.95 nm±5.28 nm δ=1.15%										
$n\left(\lambda \right)=1.947+\frac{0.5636*10^{5}}{\lambda^{2}}$										

Note: The underlined values are obtained from the transmission graph obtained by spectrophotometry; The other values are obtained from the envelope curves according to Swanepoel theory.

**Table 3 materials-14-05845-t003:** The EDX species of the 2 %Fe doped WO_3_ and pure WO_3_ layers.

Temperatures (°C)	Layer	Element	Weight %	Atomic %	Intensity	Errors %
600 °C	WO_3_ + 2%Fe	O (K)	16.86	69.86	143.59	6.14
		Fe (L)	0.19	0.23	0.62	66.09
		W (M)	82.95	29.91	266.03	3.34
680 °C		O (K)	17.08	70.29	144.76	6.16
		Fe (L)	0.03	0.03	0.09	99.99
		W (M)	82.89	29.68	264.33	3.36
600 °C	WO_3_	O (K)	17.36	70.70	77.31	6.94
		W(M)	82.64	29.30	138.36	4.09
660 °C		O (K)	17.49	70.90	76.34	6.94
		W(M)	82.51	29.10	135.21	4.04

## Data Availability

The data presented in this study are available on request from the corresponding author.

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
