# Peer review of "Influence of the Iron as a Dopant on the Refractive Index of WO_3"

_materials, 2021, doi:10.3390/ma14195845_

Round 1
Reviewer 1 Report
In this manuscript by Mariana Osiac et al., investigations of pristine WO3 and 2%Fe doped tungsten oxide 2%Fe - WO3 are reported. The samples were grown on the sapphire substrates by reactive pulsed laser deposition technique. The crystalline structure of 2, 3, and 4% iron doped WO3 showed the small shift compared with the pristine WO3.
The paper can be published in Materials. It is well-organized and written. The X-ray, SEM, refractive index and optical band gap measurements are reported in detail and provide sound observations on the changes of optical parameters with temperature and Fe concentration in the samples. These values are examined for different temperatures and amounts of Fe content.
For different temperatures in the manuscript it was found that in the presence of 2%Fe doped WO3, the nanocrystallites size varied from 60 to 190 nm. This statement from abstract is absent in conclusions and not discussed in the text of the manuscript. Figure 10 should be replotted with appropriate scales and axis captions. In Figures 8 and 9 spline plots can be added for the point sets, similar to Figure 7. Conclusions can be modified to include less numbers but more generalized observations.
The manuscript can be published in Materials with minor changes.
Reviewer 2 Report
The purpose of this paper was to evaluate the optical properties of a pure 2% (weight) Fe-doped WO3 thin film deposited by pulsed laser deposition on a sapphire substrate using the Swanpole method.
However, as in the background, WO3 is Fe-doped on the assumption that it will be used as an n-type semiconductor. Therefore, the lack of electrical characteristics is insufficient as a treatise. Experiment once after rejecting, and proceed to submit again.
Reviewer 3 Report
In this study, Influence of the Iron is studied as a dopant on the refractive index of WO3. But there are questions related to the choice of the object of research and the discussion of the results of the experiment. I recommend publication of this article after some revisions and satisfactory answers to comments.
- “The aim of this paper is to measure refractive indexes in the 300-1200nm spectral range of pure and 2%(weight) Fe doped WO3 thin films…” Why 2%(weight) Fe doped WO3 is selected? Specify the advantages over other concentrations of the dopant, justify the choice of Fe. “The pure and 2% Fe doped WO3 thin films were deposited on sapphire substrates…” – as stated in Materials and Methods, but “The effect of Fe as a dopant is visible for 3% and 4%Fe doped WO3 layers” – in Results and Discussion and in Conclusions. Where do the two more concentrations come from?
- Why the 300-1200nm spectral range is selected? It’s not discussed in the Introduction.
- The article is overloaded with tables and figures of the same type, which complicates the perception of the results. It is necessary to present the results of the experiment in a more understandable form, for example, to make combined figures and think about the need for so many tables.
Round 2
Reviewer 2 Report
It has been modified and can be accepted.
However, please describe why the electrical characteristics cannot be evaluated. Since a sapphire substrate is used, it is considered possible to evaluate the I-V characteristics.
